# Of Cockroaches and Symbionts: Recent Advances in the Characterization of the Relationship between *Blattella germanica* and Its Dual Symbiotic System

**DOI:** 10.3390/life12020290

**Published:** 2022-02-15

**Authors:** Amparo Latorre, Rebeca Domínguez-Santos, Carlos García-Ferris, Rosario Gil

**Affiliations:** 1Institute for Integrative Systems Biology (I2SysBio), Universitat de València/CSIC, Calle Catedrático Agustín Escardino, 9, 46980 Valencia, Spain; rebeca.dominguez@uv.es (R.D.-S.); carlos.garcia.ferris@uv.es (C.G.-F.); 2Departament de Genètica, Universitat de València, Calle Dr. Moliner, 50, 46100 Valencia, Spain; 3Área de Genómica y Salud, Fundación para el Fomento de la Investigación Sanitaria y Biomédica de la Comunidad Valenciana (FISABIO), Avenida de Cataluña 21, 46020 Valencia, Spain; 4Departament de Bioquímica i Biologia Molecular, Universitat de València, Calle Dr. Moliner, 50, 46100 Valencia, Spain

**Keywords:** *Blattella germanica*, model insect, symbiosis, *Blattabacterium*, gut microbiota, systems biology, resistome, antimicrobial peptides

## Abstract

Mutualistic stable symbioses are widespread in all groups of eukaryotes, especially in insects, where symbionts have played an essential role in their evolution. Many insects live in obligate relationship with different ecto- and endosymbiotic bacteria, which are needed to maintain their hosts’ fitness in their natural environment, to the point of even relying on them for survival. The case of cockroaches (Blattodea) is paradigmatic, as both symbiotic systems coexist in the same organism in two separated compartments: an intracellular endosymbiont (*Blattabacterium*) inside bacteriocytes located in the fat body, and a rich and complex microbiota in the hindgut. The German cockroach *Blattella germanica* is a good model for the study of symbiotic interactions, as it can be maintained in the laboratory in controlled populations, allowing the perturbations of the two symbiotic systems in order to study the communication and integration of the tripartite organization of the host–endosymbiont–microbiota, and to evaluate the role of symbiotic antimicrobial peptides (AMPs) in host control over their symbionts. The importance of cockroaches as reservoirs and transmission vectors of antibiotic resistance sequences, and their putative interest to search for AMPs to deal with the problem, is also discussed.

## 1. Introduction

### 1.1. The Impact of Symbiosis with Bacteria in Eukaryotic Evolution

Symbiotic relationships are widespread in nature. They can range along a continuum from parasitic to mutualistic, and a wide array of players can also be involved and contribute to phenotypes at all levels of biological organization. At present, there is no doubt that mutualistic stable symbioses have evolved independently many times in all groups of eukaryotes, contributing to their phenotypes and allowing them to explore a diversity of niches, which have had a significant impact on animal evolution [1,2].

Two main cases of symbiosis are distinguished regarding their location in relation with the host: endosymbiosis, in which the symbiont (mostly bacteria) live intracellularly inside specialized host cells (the bacteriocytes) and which normally involves one-to-one or few-to-one symbiont–host relationships [3], and ectosymbiosis, in which the symbionts live extracellularly in different organs of the host and with a many-to-one relationship involving a high number of prokaryotes (mostly bacteria, but also eukaryotes (fungi and some protists)) [4]. Endosymbionts are vertically transmitted, and they have undergone genomic and functional changes to complement the host in many respects, without being recognized as infectious agents [5]. On the other hand, ectosymbionts, which usually are horizontally transmitted, contribute to the performance of some host functions, depending on the organism or the specific body part within the organism. In the last two decades, the use of NGS technologies has revealed that many animals, from insects to mammals, possess a complex gut microbiota that plays a range of essential symbiotic roles, such as regulating the host’s metabolism and participating in nutrient uptake, the digestive process, detoxification, physiology and immunity, or protecting against pathogens colonization [6,7,8]. It is still an unsolved question as to what determines the evolutionary path towards endosymbiosis or ectosymbiosis.

### 1.2. Bacterial Symbiosis in Insects

Insects represent around 85% of animal diversity. Practically all insects are involved in some kind of symbiotic association with bacteria [9], which, at least in 15% of cases, involves a mutualistic relationship, being one of the key factors of their evolutionary success [10,11]. In most studied cases of mutualistic symbiosis, the specialized host diet is a key feature, and the lacking nutrients must be supplemented by the endosymbionts. The characterization in recent years of many endosymbiont genomes belonging to a great variety of insect clades allowed a better understanding of their roles as nutrient suppliers to cope with the deficiencies in the respective hosts’ diets (reviewed in [1,12,13]). As examples, aphids, psyllids, white flies and mealybugs feed on phloem sap, which is deficient in nitrogen compounds. All these insects harbour different endosymbionts that provide the lacking nutrients. Sharpshooters feed on xylem, which is poor in nutrients, containing mostly minerals and inorganic compounds. Tsetse flies feed on blood, which is deficient in vitamins that are also supplied by the endosymbiont. Studies performed on these insects that rely on obligate mutualistic endosymbionts to complement their nutrient-poor diets indicate that, in most cases, they depend on one or a few intracellular primary (obligate) symbionts, although they can also harbour secondary (facultative) ones, but lack a complex extracellular gut microbiota. Interestingly, some omnivorous insects have also established endosymbiotic relationships with bacteria. This is the case of cockroaches and carpenter ants, which feed on complex diets, but possess obligate mutualistic endosymbionts that are involved in nitrogen recycling and upgrading [14,15,16].

Regarding the extracellular gut microbiota, complex microbial communities inhabiting the digestive tracts have also been studied in a wide variety of insects, such as termites [17,18,19], cockroaches, crickets, beetles [20,21], lepidopterans [22,23], hymenopterans [24] and *Drosophila* [25]. These studies revealed that the nature, diversity, complexity and functions of the gut microbiota are highly variable across the different groups of class Insecta [7].

### 1.3. Symbionts Must Be Present in Every Generation

Symbiont transmission is essential to ensure the maintenance of symbiotic systems through host generations. Consequently, the means of transmission is a key element to understanding the evolution of the symbiotic microbiota. Two fundamental modes of transmission can be distinguished, namely, vertical (colony-to-offspring transfer) and horizontal (colony-to-colony transfer), which can be either directly via contact among individuals or indirectly via uptake from the environment [25,26]. Normally, free-living bacteria evolve towards a strictly vertical transmission mode when they become endosymbionts to ensure their presence in the next generation, as it is essential [11,27]. In fact, the only moment in which they are found outside the bacteriocytes is when they need to be transferred to the oocytes [28] or early embryos in parthenogenetic insects, such as some aphid lineages [29]. Ectosymbionts, on the other hand, have a less clear means of transmission, which varies in different organisms. While horizontal transfer is considered to be the general route for the acquisition of gut bacteria in mammals [30], horizontal, vertical and even mixed ways have been described in insects [22,31]. In social insects, such as termites and wood feeding cockroaches, the development of proctodeal trophallaxis, a sophisticated social behaviour in which nest-mates share droplets of hindgut contents, would ensure a deterministic transmission of their complex gut microbiota [17,32]. In addition, the environmental uptake of transient associates appears to be likely in other insects [22,33]. In some orders, such as Heteroptera and Diptera, extracellular bacteria are attached to the egg surface by the females through their own secretions or faeces, and they are later acquired by the hatching nymphs [34,35]. In Lepidoptera, the detection of core bacterial gut associates suggests a potential vertical transmission, but a consistent horizontal acquisition of gut symbionts has also been described in some species [22]. Similarly, a thorough recent study examining the gut microbiota of 94 termite species [36] concluded that a mixed mode of transmission that combines vertical and horizontal transfer occurs. However, in non-social cockroaches, even though diet can influence the final outcome, coprophagy is the main factor responsible for the transmission [37,38,39,40,41,42].

Not only is transmission important, but so is the way the different bacterial species are assembled and become established along the host development (microbiota ontogeny). Humans, for example, are not microorganism-free at birth [43], but the different bacterial species that colonize us progressively settle following a dynamic typical to an ecological succession [44,45]. Less information has been obtained about the assemblage process in insects, although it has been increasing in recent years (e.g., [26,46,47]). As for cockroaches, some model species have been used to obtain germ-free and gnotobiotic insects in the laboratory to study the way the microbiota is acquired and restored after perturbations, as will be presented in Section 3.1.

## 2. The Dual Symbiotic System in Cockroaches

Cockroaches (order Blattodea) represent an ancient lineage that emerged over 300 million years ago. They are paradigmatic of both the symbiotic systems mentioned above that coexist in each individual in separated compartments. On the one hand, similar to insects that feed on specialized diets, they present with the endosymbiont *Blattabacterium cuenotti* (hereafter *Blattabacterium*) inside bacteriocytes located in the fat body of omnivorous cockroaches and in the wood-feeding *Cryptocercus* and the giant termite *Mastotermes*. On the other hand, and similar to other omnivorous animals, they harbour in their hindgut (the posterior part of the gut) a complex microbiota. The complete genome of at least 70 *Blattabacterium* strains have been sequenced, and their metabolisms inferred, confirming their monophyletic origin in all cockroaches as well as their essentiality in hosts’ physiology [48,49,50,51]. Besides, the gut microbiota has also been characterized in several species, both at taxonomic and functional levels. Our model system is the German cockroach *Blattella germanica*, one of the few cockroaches whose microbiota has been studied using a metagenomics approach (Table 1).

### 2.1. The Essential Role of Blattabacterium

Three cell types are present in the fat body of cockroaches: adipocytes, the primary components of this tissue, specialized in storing energy as fat; uricocytes, where the surplus nitrogen is stored as uric acid, and bacteriocytes, where the endosymbiont *Blattabacterium* is located [68]. When the genome of *Blattabacterium* from *B. germanica* was sequenced, its role in the synthesis of essential amino acids was inferred [15]. However, the most striking result was to find that it has retained the complete urea cycle, plus the two genes that encode the catalytic core of the urease (*ure*AB and *ure*C; EC 3.5.1.5). In fact, all *Blattabacterium* strains from cockroaches sequenced so far possess a complete (or almost complete) urea cycle and an active urease that converts urea into ammonium; only those strains from *Cryptocercus punctulatus* and *Mastotermes darwiniensis* have the urea cycle interrupted by the absence of argininosuccinate lyase (EC 4.3.2.1) [69]. The proposed metabolic pathway for the mobilization of the uric acid from the neighbour uricocytes is as follows: uric acid would be degraded into urea by the successive action of three host enzymes: urate oxidase (uricase; EC 1.7.3.3), allantoinase (EC 3.5.2.5) and allantoicase (EC 3.5.3.4). Then, urea is degraded into ammonium and CO_2_ by the endosymbiotic urease. Finally, ammonium can be incorporated into organic compounds via the synthesis of glutamine by the host enzyme glutamine synthetase (EC 6.3.1.2) [16], or into glutamate by the endosymbiont enzyme glutamate dehydrogenase (EC 1.4.1.2) [70,71], which can be used by the endosymbiont as a source of organic nitrogen for the synthesis of essential amino acids. Using transcriptomic analysis, we identified in the fat body transcripts corresponding to all the genes necessary for the transformation of uric acid into urea, although the corresponding genes compose a metabolic patchwork between the bacteria and host [16]. Therefore, *Blattabacterium* plays an essential role in insect nutrition by participating in nitrogen recycling and supplying essential amino acids to the host. Recently, a comparative genome analysis of 67 *Blattabacterium* strains revealed a massive parallel gene loss. Thus, 200 out of the 566 protein-coding genes under study were lost in at least one lineage, while 25 were lost independently from 10 to 24 times. Such losses affect the genes related to the biosynthesis of amino acids and cofactors to a greater extent, while those involved in nitrogen metabolism are retained [49].

### 2.2. Termites Have Lost Blattabacterium

Termites (Blattoidea: Isoptera) are the closest relatives to cockroaches (Blattoidea: Blattaria). In fact, they are considered to be social cockroaches [72], and form a monophyletic clade that evolved from wood-feeding cockroaches about 170 Mya, sharing a common ancestor with the subsocial cockroach genus *Cryptocercus* [73]. The acquisition of *Blattabacterium* took place in a common ancestor of cockroaches and termites [74,75]. However, termites have lost *Blattabacterium*, with the single exception of the xylophagous lower termite *M. darwiniensis* [76,77], which has preserved the endosymbiont, but with a reduced genome, similar to that of *Cryptocercus* [67,74,78]. Key events in the evolution from the common ancestor of termites and Cryptoceridae allowed the shift from an omnivorous to a cellulose-based diet [79]. The acquisition of cellulolytic flagellates and a specialized gut microbiota were essential to metabolize cellulose and upgrade such a nitrogen-poor diet [80]. In higher termites, the loss of flagellates and their endosymbionts resulted in a fully prokaryotic gut microbial community, which is involved in lignocellulose degradation, but which also plays an important role in the fixation, recycling and upgrading of nitrogen, allowing the replacement of the lost *Blattabacterium* [81]. The complex gut microbiota of higher termites has been scrutinized in recent years, through 16S rDNA and metagenomic studies, including MAG (metagenome-assembled genome) sequencing [82,83]. These works confirmed that the gut microbiota composition in termites is highly variable depending on host taxonomy, host diet and the microenvironment within the host, and that different bacterial players can be involved in different termite groups, even within the same or similar feeding habits.

### 2.3. The Gut Microbiota of Cockroaches and Its Not Fully Untangled Role

Contrary to the well-understood role of the endosymbiont in the fat body of the cockroaches, the role of the hindgut microbiota in the host physiology and its putative crosstalk with the endosymbiont have not yet been completely deciphered. In recent years, studies focused on specific phenotypes influenced by the gut microbiota in *B. germanica* have been carried out, evidencing a role of faecal agents in cockroaches’ aggregation behaviour [84], or a protective effect of the microbiota against exogenous bacterial pathogens [85], just to mention some examples. Nevertheless, most approaches to the role of the gut microbiota are based on in silico predictions. To understand why *Blattabacterium* has not been replaced by the gut ectosymbiotic community in cockroaches, as has occurred in termites, the diversity, complexity, functions and developmental dynamics need to be deeply studied. In recent years, the gut microbiota of different species of roaches has been published, mainly using a 16S rDNA barcoding approach (see Table 1), revealing the complexity of the bacterial composition. Following the recently validated names for the prokaryotic phyla by the International Committee on Systematics of Prokaryotes (ICSP) [86], the most abundant phyla are Bacteroidota, Bacillota (formerly Firmicutes, shared with other omnivorous animals including humans), Pseudomonadota (formerly Proteobacteria), and Fusobacteriota (shared by many insects). In addition, a metagenomic study on *B. germanica* revealed the presence of some species from other phyla, including members of the Actinomycetota (formerly Actinobacteria), Spirochaetota, Synergistota and Verrucomicrobiota, as part of the core gut associates [38]. Since the most abundant phyla are present in both termites and cockroaches, it would be necessary to look at minority representatives to solve the mystery of why cockroaches still need *Blattabacterium*.

The bacterial composition of the cockroach hindgut microbiota depends on an intricate interaction between host genetics and environment. On the one hand, the host’s species play a key role in shaping it. In omnivorous gregarious cockroaches, there is a strong association between host species and gut microbiota composition, and even a weak but significant correlation between host family and gut microbiota [52], while different species living in sympatry have a species-dependent composition [60]. On the other hand, insects from the same species living in different environments host different bacterial communities in their guts [20,55]. Thus, the composition is more diverse in wild cockroaches than in lab-reared ones [60,65,87], a change that appears to be associated with their omnivorous diet, which can be highly variable for wild roaches.

As for the changes in composition during the developmental stages, the core bacterial components of the microbiota are acquired early in the cockroach’s life [64]. The bacterial diversity increased by two orders of magnitude from the first to the second nymphal stage, when 86% of the total detected bacterial taxa appeared, and remained similar between the third nymphal and adult stages, although the cooperative bacterial networks are not completely defined until reaching the adult stage [39]. Furthermore, the relative abundance of the taxa changes along with development, and an ecological succession throughout the insect life cycle can be found, i.e., *Fusobacterium* accumulated with aging, while *Bacteroides* decreased [65].

In the absence of empirical evidence for most proposed roles, metagenomic approaches followed in lab-reared populations of *B. germanica* allow us to infer the putative symbiotic functions carried out by the gut microbiota in this insect [38,39]. Most detected taxa, as well as the specialized functions they perform, could be related with their omnivorous diet. In our studies, we found a wide repertoire of genes involved in metabolic processes, being especially abundant those participating in the metabolism of proteins and carbohydrates, but also pathways related to energy production, the synthesis of vitamins and nitrogen metabolism. A striking result is the significant number of antimicrobial resistance genes (ARGs) and genes related to mobile and extrachromosomal element functions that were detected, which deserves a specific section in this review.

## 3. Why *B. germanica* has Preserved Both Symbiotic Systems?

At this point, it is not clear yet why B. germanica needs both the endosymbiont and the complex gut ectosymbiotic community. It could be possible that (i) there is crosstalk between the two systems in the adult insect, (ii) that each one of the systems is needed in a differential way through the developmental stages, or (iii) that somehow the gut microbiota might complement (and maybe replace) *Blattabacterium*. A good way to address these questions is to cause and analyse the changes that occur in both symbiotic systems under the effect of different perturbations, such as different diets or antibiotic treatments, compared to control conditions.

### 3.1. Perturbing the Gut Microbiota to Learn about It

An approach that can help to understand the possible crosstalk between endo- and ectosymbionts is to feed them on modified diets. Perez-Cobas et al. (2015) [65] used lab-reared populations of the same genetic line of *B. germanica*, but them fed with different diets containing different amounts of protein (0, 25 and 50%). However, any essential function that the microbiota performs must be preserved, even in these conditions. The results showed that the gut microbiota of *B. germanica* is highly dynamic, as the bacterial composition was reassembled in a diet-specific manner over a short period of time, with the no-protein diet promoting higher diversity. This was a surprising result, since in *P. americana*, a highly stable core microbial community has been observed, with highly similar composition between individuals and very small variations in response to highly diverse solid diets [58]. This discrepancy could be due not only to a species-specific effect, but also to the diet composition, which is very different in both studies.

Another approach to learn about the functions and dynamics of the hindgut microbiota consists of rearing germ-free cockroaches (lacking microbiota) and introducing controlled microbial communities (gnotobiotic cockroaches). Because nymphs are free of hindgut microbiota at the time of ootheca hatching, this can be achieved by sterilizing the surface of the ootheca before this event [84,88]. After that, the colonization of different controlled microbial communities can be easily accomplished through the administration of defined microbial combinations or faeces from control populations of the same or other cockroach species, as well as from other animals (termites, mice). These studies have been performed in different laboratories on several model cockroaches, such as *S. lateralis* [42,89], *P. americana* [41,90] and some leaf litter-feeding roaches [53], and have been used for studying both the microbiota dynamics and its functions.

Another way to introduce perturbations on both symbiotic systems is the use of antibiotics that could have a differential effect on them. Figure 1 represents a typical experiment in our laboratory to perform multi-omics studies on these perturbed *B. germanica* populations. The populations can be analysed through the different developmental stages (Figure 1a). Following the scheme of Figure 1b, the selective loss of gut bacteria has been achieved using the antibiotics ampicillin and rifampicin (against Gram-positive and some Gram-negative bacteria), kanamycin (against Gram-negative bacteria) and vancomycin (against Gram-positive bacteria). Only rifampicin has been proven to affect *Blattabacterium*, but all of them were expected to affect the fitness of *B. germanica* to a greater or lesser extent [38,39].

We also checked if the microbiota composition recovered after a perturbation (i.e., if it is resilient), by monitoring how the microbial community composition recovered after the treatment cessation. The effect on the microbiota recovery of adding faeces to the diet has also been evaluated. According to our empirical observations, the composition of the bacteria changes quite rapidly, depending on the action of the specific antibiotic, while the main roles were conserved, even in the antibiotic-treated samples, suggesting a buffering capacity of the microbiota to perform essential symbiotic functions [38,39]. Thus, although the bacterial species are different depending on the treatment, functional redundancy must be essential to guarantee the function of the system after being exposed to any kind of perturbation. When the antibiotics are removed, the composition and structure of the microbiota is quickly recovered in the next generation, and such recovery is faster if faeces are provided with the diet.

### 3.2. The Resistome

The set of all types of antibiotic resistance genes (ARG) identified in a microbial community is known as the resistome [91]. The evolution and widespread distribution of ARGs is causing a continued increase in multidrug-resistant bacteria, which is having a dramatic impact on infection control, as diseases that were once easily treatable are becoming deadly again [92]. For this reason, monitoring the resistome in different environments from the One Health (human–animal–environment) perspective is essential for a comprehensive understanding of the origin, emergence, dissemination and evolution of ARGs [93].

Besides areas of clinical activities, the highly significant relevance of natural environments as reservoirs for antibiotics and their corresponding ARGs has been highlighted by the availability of metagenomics data [94,95,96]. The study of several gut microbiomes (including not only the microorganisms composing the microbiota, but also the viruses and the community “theatre of activity” [97]) revealed that insect pests can represent a public health problem as reservoirs of ARGs [98,99]. This is the case of our model system *B. germanica*, a cosmopolitan cockroach that lives in close contact with humans and it is found in houses, hospitals and in unsanitary and insalubrious areas. For this reason, the presence of ARGs that can be carried and transmitted by urban cockroach species have been studied through culture-based techniques in the presence of antibiotics in order to isolate multidrug resistance bacteria [100,101]. However, as most components of the gut microbiota cannot be cultured, the study of metagenomes could provide valuable information on this subject. The resistome of *B. germanica* has been recently determined by combining hindgut metagenomics data from lab control populations and treating them with three antibiotics with a different spectrum of activity and mechanism of action: the beta-lactam ampicillin, the aminoglycoside kanamycin, and the glycopeptide vancomycin [39]. The study confirmed that ARGs that confer resistance to several broad-spectrum antibiotics frequently used in clinics, agriculture and farming (i.e., beta-lactams, folate synthesis inhibitors, tetracyclines, amphenicols, glycopeptides, polymyxins and aminoglycosides) are present on this cockroach microbiome, even without any antibiotic selective pressure, an indication that this species is a natural reservoir of ARGs. However, similarly to what has been found in studies of other gut microbiomes (humans included), the relative abundance of ARGs increased significantly after an antimicrobial treatment, leading to antibiotic-specific profiles [102,103]. Furthermore, a single course of antibiotic treatment apparently causes a permanent increase in the presence of ARGs, since their high relative abundance remains in the absence of treatment, similarly to what has already been described in other insect pests, such as mosquitoes [99]. Moreover, many mobile element-related components were also detected, mainly DNA transposons, which are known carriers of antibiotic resistance [104]. The presence of genetic elements involved in DNA mobilization indicates that they can be transferred among microbial community partners, participating in resistance transmission. Therefore, cockroaches can be considered to be both reservoirs and potential transmission vectors of ARGs.

### 3.3. Silencing Blattabacterium

A good way to ascertain whether the gut microbiota can replace *Blattabacterium* functions is to obtain aposymbiotic populations and to determine the effect of the removal of the endosymbiotic populations on the cockroaches’ fitness. Several approaches to produce aposymbiotic cockroaches were tested in early symbiotic studies (with high temperature, treatment with lysozyme and the use of antibiotics) [105,106]. However, in these studies, only one generation was analysed for the presence of the endosymbiont, or the treatment pressure was maintained over several generations, an indication that at least a reduced amount of *Blattabacterium* was able to reach the next generation after treatment cessation. A study performed on *P. americana* showed that cockroaches kept under starvation conditions end up dying without mobilizing the endosymbiont population to obtain energy and nutrients for subsistence [68]. Therefore, as *Blattabacterium* has been proven to be essential for host survival, it is not possible to generate a stable aposymbiotic population, and the only option is to reduce as much as possible the amount of the endosymbiont, thus obtaining quasi-aposymbiotic insects. It was previously found that the broad-spectrum antibiotic rifampicin is able to reduce by several orders of magnitude the presence of *Blattabacterium* in the next generation when supplied to females throughout the adult stage [37]. In the second generation after rifampicin treatment, the microbiota in adults was almost fully recovered, despite the fact that the amount of *Blattabacterium* was five or six orders of magnitude lower than that of a normal population, which would indicate that the composition of the microbiota was not affected by the reduction in the endosymbiont load. Because the antibiotic also affects the rifampicin-sensitive gut bacteria and can affect the insect viability, in another experiment, designed to progressively reduce the *Blattabacterium* load while affecting the microbiota as little as possible, the same antibiotic was provided only during the window time for oocyte infection in the adult stage and, then, the microbiota was allowed to recover before sampling [107]. It was found via qPCR that the amount of *Blattabacterium* was highly reduced in the second generation after two successive treatments in some individuals, in all of which the composition of the gut microbiota was similar to that of the controls. However, that apparently normal microbiota was not able to compensate for the endosymbiont role because the host’s fitness was drastically affected, as shown by different parameters: development (number of days from hatching to reach adult stage and to the ootheca hatching), reproductive capability (number of nymphs per ootheca) and viability (% nymphs’ survival in G2). Thus, the gut microbiota is not affected by the strong reduction of endosymbiont, and a normal gut microbiota appears to be unable to replace the functions of *Blattabacterium*. Nevertheless, this observation needs to be confirmed, as the detected effect was not homogeneous throughout the cockroach population, and due to the high mortality caused by the endosymbiont load reduction.

## 4. Role of the Host Immune System in the Symbiotic Interaction

To understand the whole symbiotic system, it is also necessary to evaluate the engagement of the host in controlling the interaction with both *Blattabacterium* and the gut microbiota. While some new evolutionary innovations specifically related to the symbiosis must be involved, it has been proposed that mutualistic symbiotic interactions are mainly controlled by the same weapons used against pathogens [108], including the innate immune system [109]. Antimicrobial peptides (AMPs) are well-known effectors of the innate immune response in plants and animals involved in the defence against infections, but some of these peptides are also produced by the host during its mutualistic interaction with bacteria [110]. It has been proposed that the innate immune system, with its host-specific AMPs, evolved in early branching metazoans to control beneficial microbes and not pathogenic bacteria [111]. Even though many questions remain open, recent advances in the identification of factors involved in host–symbiont communication and integration for a number of symbiotic associations have revealed the important role played by AMPs and AMP-like factors, the “symbiotic” AMPs [112,113]. The aim of these symbiotic AMPs would not be killing, but keeping the bacterial population under control [110,114].

In this context, it has been proposed that specific AMPs would be involved in the permeation of bacterial membranes, allowing the transient flux of metabolites across the host–symbiont interface without compromising the endosymbiont integrity, as described in vitro in *Alnus*–*Frankia* symbiosis [115]. This would explain the traffic of metabolites through the membranes of *Blattabacterium*, where the genome has undergone a genomic reduction syndrome [13], and the extremely low gene set coding for transporters cannot account for the transport of all necessary metabolites [15]. Moreover, some of these symbiotic AMPs could act intracellularly, controlling the bacterial metabolism and its physiology, and targeting specific components of the cellular machinery involved, for example, in cytokinesis or translation [112,116], which would help to keep the endosymbiont population under control.

AMPs also appear to participate in the control of host–symbiont interaction at the level of the gut microbiota. The composition of the gut microbial community is affected by environmental factors, but it is also defined by host genetic factors, specifically through the action of the AMPs in the gut, which act in the selection of the components of the microbiota. It has recently been demonstrated, through the mutation of the complete set of AMP genes in *Drosophila* using CRISPR/CAS9, that AMP expression in the gut is directly involved in the structure and abundance of the microbial community that colonizes the hindgut [108].

Due to these considerations, knowing the full complement of AMPs of the eukaryotic hosts that participate in symbiotic associations is of great interest. Although they are not complete yet, the genome projects of *B. germanica* [117] and *P. americana* [118] revealed that their repertoire of coding genes is larger than that of other related insects due to gene families’ expansion, including some related to microbial defence and immune response. By integrating genomic and transcriptomic data, our group described 39 AMP-coding genes in *B. germanica* [119]. They belong to five families: defensins, termicins, drosomycins, attacins and blattellicins. The latter emerged recently after the duplication and fast evolution of an attacin gene, which is now encoding larger proteins with the presence of a long stretch of glutamines and glutamic acids (Glx-rich region). Interestingly, these genes are an evolutionary innovation in the *B. germanica* lineage, since a single gene with similar characteristics, but not the Glx-rich segment, has been identified in a species from a closely related genus.

A screening of AMP gene expression in *B. germanica* transcriptomic projects available in the NCBI database, corresponding to 28 whole-body samples from different developmental stages, tissues and body parts, confirmed that some AMPs are expressed specifically in some developmental stage [119]. In fact, the expression of the novel blattellicins is restricted to adult females, although the tissue in which they are expressed is unknown. It will be interesting to check if any of these AMP genes are specifically expressed in the symbiotic organs, fat body or gut, and could be involved in the control of the symbiotic associations. Finally, it cannot be ruled out that some cryptic symbiotic AMP-like factors (unnoticed in the analyses focused on canonical AMPs) could be involved in the endosymbiotic interaction of *B. germanica* with *Blattabacterium*, as it has been proposed in the aphid–*Buchnera* endosymbiosis [120,121], where some bacteriocyte-specific cysteine-rich peptides, which exhibit in vitro antimicrobial activity with increased membrane permeability, could participate in endosymbiont control.

## 5. Conclusions and Perspectives

Cockroaches are omnivorous animals that, contrary to other animals, maintain two complementary symbiotic systems: the endosymbiont *Blattabacterium* that upgrades the nitrogen metabolism of the host across the developmental stages, including times in which no nutrients are provided through the diet, and a rich and complex hindgut microbiota which, despite its composition and structure along development being well known, is still under study regarding its function in complementing the host. For a better understanding of these functions and a possible interplay of both symbiotic systems, in recent years, many efforts have been made to detect what happens when they are exposed to different perturbations. These studies have revealed that the gut microbiota is highly resilient, as it is quickly recovered after the perturbation ceases. Additionally, the results obtained until now anticipate that the gut microbiota members are good candidates to participate in nutritional and defence functions. Key to understanding the putative crosstalk between the three members of the symbiotic association will be to perform perturbations on gnotobiotic and quasi-aposymbiotic populations and compare them to control conditions. These experiments are currently being carried out in our laboratory.

Recent studies have focused on the role of AMPs in some host–symbiont associations, and it would be interesting to evaluate if some symbiotic AMPs participate in the tripartite crosstalk in *Blattella germanica*. In this context, the study of the implications of short non-coding RNAs (both from the host and the symbiont) is becoming an exciting new line of research [122,123]. From an applied point of view, AMPs are also a plausible alternative to antibiotics. Insect pests (cockroaches included) can act as reservoirs of antibiotic resistance genes (ARG) and also participate in their spread, but they are also an important source of antimicrobial peptides (AMPs), which supersede antibiotics in our battle against pathogens. Their arsenal of AMPs can be explored to help fight antibiotic multi-resistance in bacteria.

## Figures and Tables

**Figure 1 life-12-00290-f001:**
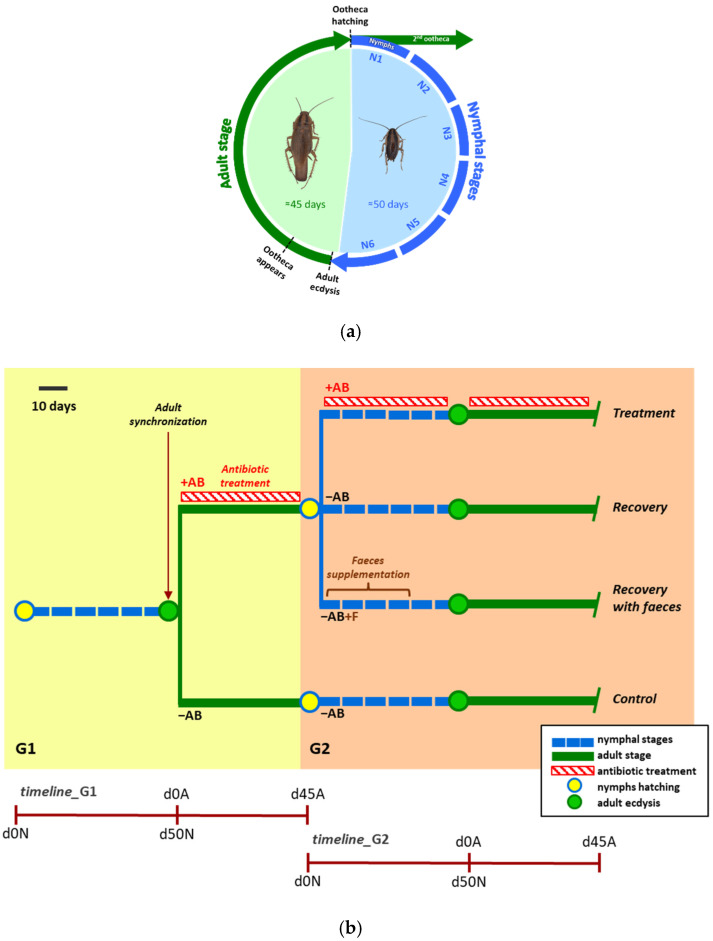
Life cycle of *B. germanica* and experimental design for antibiotic treatment and recovery. (**a**) The life cycle of *B. germanica* consists of three developmental stages: egg, nymph (N) and adult (A), and is completed in approximately 95 days at 26 °C. The egg begins its development inside the ootheca (egg case), which contains around 30–40 eggs and remains attached to the female until the nymphs are born. After approximately 35 days, the nymphs emerge from the ootheca (ootheca hatching, d0N). Nymphs moult and go through 5 or 6 nymphal stages (N1 through N6), shedding their exoskeleton and growing, and reach full maturity in approximately 50 days (d50N). After the final moulting (adult ecdysis), the sexually active adult emerges (stage with wings, d0A). Adults breed immediately, and the ootheca with fertilized eggs emerge in the female in just one week. (**b**) A typical experiment starts with a synchronized adult population, composed of individuals collected between 0 and 48 h after adult ecdysis in generation 1 (G1). Then, the population is divided into two subpopulations: the control one is never treated with antibiotics (−AB), until reaching the second generation (G2); the experimental one is treated with antibiotics (+AB) during the adult stage. The antibiotics are removed when the ootheca is fully formed (a few days before hatching), and newly emerged nymphs are divided into three subpopulations in G2: the first one is immediately treated with antibiotics (+AB) during the complete life cycle (Treatment), the second one is kept without any further antibiotic treatment (−AB; Recovery) and the third one is kept without any further antibiotic treatment, but is supplemented with faeces obtained from a control population that has been never treated with the antibiotic (−AB+F; Recovery with faeces).

**Table 1 life-12-00290-t001:** Blattodea species with dual symbiotic systems whose gut microbiota has been studied using 16S barcoding and metagenomics approaches (in bold). The table does not intend to be exhaustive.

Host Scientific Name	Family	Reference
*Blaberus craniifer*	Blaberidae	[52]
*Byrsotria rothi*	Blaberidae	[53]
*Diploptera punctata*	Blaberidae	[52,54,55]
*Elliptorhina chopardi*	Blaberidae	[55]
*Eublaberus posticus*	Blaberidae	[55]
*Gromphadorhina portentosa*	Blaberidae	[52]
*Lucihormetica verrucosa*	Blaberidae	[52]
*Nauphoeta cinerea*	Blaberidae	[52]
*Opisthoplatia orientalis*	Blaberidae	[55]
*Oxyhaloa deusta*	Blaberidae	[52]
*Panchlora* sp.	Blaberidae	[**[56]**]
*Panchlora viridis*	Blaberidae	[52]
*Panesthia angustipennis*	Blaberidae	[55,57]
*Pycnoscelus surinamensis*	Blaberidae	[52]
*Salganea esakii*	Blaberidae	[55]
*Schultesia lampyridiformis*	Blaberidae	[52,55]
*Blatta orientalis*	Blattidae	[55]
*Eurycotis floridana*	Blattidae	[55]
*Periplaneta americana*	Blattidae	[52,58,59,60]
*Periplaneta fuliginosa*	Blattidae	[52,59,60]
*Periplaneta japonica*	Blattidae	[59]
*Shelfordella lateralis*	Blattidae	[42,61,62]
*Ergaula pilosa*	Corydiidae	[52]
*Ergaula capucina*	Corydiidae	[52,55]
*Polyphaga aegyptiaca*	Corydiidae	[52]
*Therea olegrandjeani*	Corydiidae	[52]
*Cryptocercus punctulatus*	Cryptocercidae	[55,63]
*Blattella germanica*	Ectobiidae	[37], [**[38]**,**[39]**], [64,65], [**[66]**]
*Parcoblatta fulvescens*	Ectobiidae	[52]
*Symploce macroptera*	Ectobiidae	[55]
*Symploce pallens*	Ectobiidae	[52]
*Rhyparobia maderae*	Nauphoetidae	[55]
*Mastotermes darwiniensis*	Mastotermitidae	[55,67]

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
