# Peer review of "Of Cockroaches and Symbionts: Recent Advances in the Characterization of the Relationship between Blattella germanica and Its Dual Symbiotic System"

_life, 2022, doi:10.3390/life12020290_

Round 1

Reviewer 1 Report

This is a well-written review paper, which primarily focused on microbial symbiosis in cockroaches. Nevertheless, I have several concerns given below, which are minor but essential. These issues should be amended or confirmed before publication.

Abstract

  • L14: The word “were” should be replaced with “where” or something appropriate.

Main text

  • L31-L53: It appears that the authors assume that ecto(cyto)symbionts comprise just the gut symbionts. However, some insects harbor yeast-like symbionts in the body cavity (or hemolymph), which inherit vertically (1,2). Some insects including attine ants and fungus-growing termites have established an external symbiotic relationship with basidiomycete fungi (3), which is also a sort of ectosymbionts. Such examples should be taken into consideration and revise the introduction appropriately.
  • L55-L57: I am skeptical that only 15% of insects are associated with bacterial symbionts, and 85% of them are free form them. Even Wolbachia are thought to infect more than half of all insect species (4). Please check this sentence and cite appropriate references (such as Buchner’s book or more recent literatures) for more accuracy.
  • L75-78: Honey bees are obviously good examples as well (5).
  • L115-L118: See literatures (6,7), which are also good examples for the assemblage process in Shelfordella.
  • L120: Replace “and” with “an”.
  • L124-133: Italicize Blattabacterium or Blattella in this paragraph.
  • L125: Although I don’t feel it is valid enough, three new species of Blattabacterim have been proposed (8). Just for your reference.
  • L128: This is underestimated. Indeed, 67 Blattabacterium genomes have recently been sequenced in (9,10). Table 1 should also be revised accordingly.
  • Table 2: The microbiota in Byrsotria rothi has also been determined (11), while I recommend that you exclude termites (Mastotermes) from the list (and this exclusion should be indicated in the table caption). Otherwise, you should add many other termite species to this table.
  • L144-L146: Ref. 91 should be cited here.
  • L193-L196: This sentence is somewhat sketchy, because the gut microbiota in termites is fairly variable depending on the host phylogeny, the diet, and/or the gut compartment (12).
  • L201-L203: Note that the degree of genome erosion in Blattabacterium is markedly different among cockroaches, even those without gut symbionts (9,10). Perhaps, it is better to summarize some perspectives from these literatures.
  • L270: Replace Figure 1A and 1B with Figure 1a and 1b.
  • L386: Spell out AMPs as this is a first appearance.
  • L687 (Ref. 92): Do not abbreviate the journal name (i.e. Biology), which consists of a single word.

References (only doi numbers are listed)

  1. 1093/oxfordjournals.molbev.a003901
  2. 1073/pnas.1803245115
  3. 1146/annurev-ento-040920-061140
  4. 1093/ee/nvy188
  5. 1186/s40168-021-01174-y
  6. 1128/AEM.03700-15
  7. 1128/mSphere.01023-20
  8. 1016/S1055-7903(02)00330-5
  9. 1016/j.cub.2020.07.034
  10. 1093/molbev/msab159
  11. 1186/s12866-019-1601-9
  12. 1146/annurev-micro-092412-155715

Reviewer 2 Report

This review by Latorre et al. focuses on recent advances in the study of the relationship of the German cockroach with its endosymbiont and ectosymbionts. The review is well written and the scientific discussion is rigorous and accurate. However, my primary concern regarding this review is that it seems to primarily focus on and over emphasize a handful of the authors’ own publications while overlooking or not fully incorporating a number of other important and relevant studies. Thus, the review limited in its impact to the community because it is not a holistic overview of the topic.

Sections 3, 4, and 5 appear to be largely discussion of some of the author’s recent work, and it is unclear how this discussion is useful beyond rehashing what is already presented in those papers.  On the other hand, sections 1 and 2 are very general. In other words, the review very rapidly transitions from a general treatment of insect symbiosis and very brief overview of cockroach-microbe interactions to an in depth discussion of a few of the authors’ own studies.

I believe the review should be modified to incorporate recent related advances made by others by expanding sections such as 2.3, 3.1, and 3.2, as I specify below. This would significantly broaden the impact of the review, which is otherwise limited in its current form.

Some specific points:

-In general, the abstract indicates some of the topics that will be discussed in the review, but in my opinion, it needs to be extensively edited or re-written. It is not clear from the title and abstract what this review is trying to achieve and how it will be organized. Consider a title like “Recent advances in the study of the relationship of the German cockroach with its endosymbiont and ectosymbionts”

- Lines 16-19 are particularly unclear.

-Section 2.3, There are numerous functional studies that have elucidated various physiological roles for the microbiota of German cockroaches (aggregation behavior by Wada-Katsumata et al 2015, colonization resistance against pathogens like E.coli by Ray et al 2020, insecticide resistance by several investigators such as Zhang & Yang 2019), but the authors ignore those and several others here. In fact, it appears that the focus of this section, as with most of the review, is their own work only.

-Section 3.1, Consistent with the trend above, what about studies that have taken the approach of generating B. germanica lacking a gut microbiota by sterile rearing approaches as described in Wada-Katsumata et al. 2015, Mikaelyan et al. 2016, and Ray et al. 2020? This third approach is not mentioned at all but has numerous advantages.

-Section 3.2, There have been a multitude of studies that have gone beyond using metagenomics to examine the resistome of the German cockroach microbiota by doing culture based analyses to identify antimicrobial resistant microbial species associated with the insects and their determinants, but these are not mentioned. Including these studies would add relevance and further context to the metagenomics studies discussed.

Round 2

Reviewer 2 Report

No further comments.